# Measurement and Coupling Coordination of High-Quality Development in Guangdong Province of China: A Spatiotemporal Analysis

**DOI:** 10.3390/ijerph20054305

**Published:** 2023-02-28

**Authors:** Jincan Hu, Junyi Liang, Litao Tian, Shaojian Wang

**Affiliations:** 1School of Public Administration, Guangzhou University, Guangzhou 510006, China; 2Guangdong Provincial Key Laboratory of Urbanization and Geo-Simulation, School of Geography and Planning, Sun Yat-sen University, Guangzhou 510275, China

**Keywords:** high-quality development, level measurement, coupling coordination, spatial and temporal patterns, Guangdong province

## Abstract

Regional high-quality coordination plays a crucial role in promoting high-quality national development. Guangdong province is a trailblazer in China’s reform and opening-up policies and high-quality development. This study analyzes the high-quality development of the economic, social, and ecological environments from 2010 to 2019 in Guangdong with the entropy weight TOPSIS model. Meanwhile, the coupling coordination degree model is used to investigate the spatial-temporal pattern of the coupling and coordinated development of the three-dimensional system in 21 prefecture-level cities. The results show that the high-quality development index of Guangdong increased from 0.32 to 0.39, a 21.9% increase during 2010–2019. The Pearl River Delta had the highest value of the high-quality development index, whereas Western Guangdong had the lowest level in 2019. Guangzhou, Shenzhen, Zhuhai, and Dongguan are the core cities in the high-quality development of Guangdong, with the index decreasing from the Pearl River Delta estuary cities to the province’s edge. We also found that the coupling degree and coupling coordination of high-quality development of the three-dimensional system grew slowly during the study period. Half of the cities in Guangdong entered the stage of benign coupling. Except for Zhaoqing, all cities in the Pearl River Delta have a high coupling coordination degree of high-quality development of the three-dimensional system. This study provides valuable references for the high-quality, coordinated development of Guangdong province and some policy suggestions for other regions.

## 1. Introduction

The 19th CPC National Congress states that China’s economy has been transitioning from a phase of rapid growth to a stage of high-quality development [1]. It is a pivotal stage for adapting to the principal contradiction facing Chinese society and ecological conservation. The 2022 Central Economic Work Conference also emphasizes that China must make high-quality development a top priority to make China a modern socialist country in all respects. Therefore, the current resource- and investment-driven economy should be transformed into an innovation-driven economy. More efficient use of advanced factors of production, lower ecological costs, and greener technology are helping China’s economy moves towards a stage of high-quality development.

Existing studies have different understandings of high-quality development since it was proposed. Some literature focuses on a single evaluation index, such as total factor productivity [2], economic growth rate [3], or labor productivity [4,5,6], to quantify high-quality development. For instance, Beugelsdijk et al. assessed European development with the total factor productivity [7]. Others constructed a multi-dimensional evaluation index system of high-quality development [8,9,10,11]. Chen and Huo created an evaluation index system to assess the high-quality development of China’s economy, involving innovation, coordination, green, openness, and sharing development [12].

Previous studies attached great significance to high-quality economic development, whereas some of them ignored the fact that high-quality development is part of a complex organic whole involving the economy, society, culture, and ecology [13]. Although most scholars analyze the coupling coordination of just two systems, the coupling coordination of high-quality development is multi-dimensional [14,15]. The existing studies are more concerned with the coupling coordination of two systems from the province, city, basin, and city-cluster levels, such as the ecological economy environment [16,17,18,19], demographic economy [20,21,22], and industrial economy [23,24,25]. These studies pay attention to scientific and technological innovation, industrial structure, urbanization, social development, and ecological environments, whereas the coupling coordination of multiple dimensions receives little research attention. 

China’s high-quality development, which regional growth poles are responsible for pursuing, sets a higher requirement for regional coordination. Taking advantage of being adjacent to Hong Kong and Macao and having the longest coastlines, Guangdong province is a pioneer in reform and opening-up policies. As a strong economic province in China, a cascade of development issues caused by rapid urbanization and economic transformation, such as unreasonable economic structure, a significant income gap between urban and rural areas, unbalanced regional development, and ecological damage, have become common, affecting the high-quality coordination of its economic, social, and ecological environment systems. Therefore, this study attempts to build a three-dimensional (3D) system of economy, society, and ecological environments, as well as analyze the coupling coordination of these systems, aiming to promote high-quality regional development. 

The existing literature mainly focuses on building an evaluation index system, coupling coordination, and empirical analysis regarding high-quality development. Nevertheless, these studies are limited in several ways. First, many studies use the single index, such as total factor productivity, to measure high-quality development, while this substitution does not fully recognize the multi-dimensional characteristics. Second, some scholars analyze the interactions between high-quality economic development and other influencing factors, including demographic structure, environmental regulation, and urbanization. However, a comprehensive assessment of coupling coordination among economic, social, and ecological systems is important to realize high-quality development, and few studies examine the relationship among them. This study makes up for enriching the existing literature. From the scale of a prefecture-level city, this study aims to establish a comprehensive evaluation index system of the economy, society, and ecological environments. The results of this analysis depict an overview of current high-quality development in different cities in the Guangdong province, helping us design differentiated policies to promote high-quality development at the city level. Further, we seek to analyze the coupling and coordinated development of the 3D system in different regions of Guangdong province during 2010–2019 with the coupling coordination degree model. It not only helps to theoretically understand the coupling coordination mechanism among economic, social, and ecological systems to achieve sustainable development but also helps the government to enact more rational and diversified strategies for rural revitalization, the new type of urbanization, and coordinated urban and rural development.

The structure of this paper is as follows. First, we present the study area and establish evaluation indexes for high-quality development. Second, we introduce the entropy weight TOPSIS method and coupling coordination degree model to analyze the interactions among economic, social, and ecological systems. Third, explain the results and discuss our main findings. Lastly, policy implications, limitations, and suggestions for future research are provided in the Conclusions.

## 2. Materials and Methodology

### 2.1. Study Area

Guangdong province, lying between 20°09′ N–25°31′ N and 109°45′ E–117°20′ E, is situated in the southernmost part of mainland China. It borders Hong Kong and Macao Special Administrative Regions to the south, and Fujian to the east, Jiangxi and Hunan to the north, and the Zhuang Autonomous Region of Guangxi to the west (Figure 1). There are 21 prefectural-level cities with a total area of 179,725 km^2^. The northern part of Guangdong is covered with mountains and hills with a higher terrain, whereas the southern region is dominated by plains with a lower terrain. Located in the East Asian monsoon region, Guangdong has a subtropical monsoon climate with an average annual temperature of 21.8 °C and an average annual precipitation of 1789.3 mm. As the most economically developed province in China, Guangdong’s GDP has a 1.9% year-over-year growth rate, reaching 1291.19 trillion yuan in 2022. The population ranks first nationwide, with a total permanent resident population of 126.24 million in the seventh national census in 2020.

### 2.2. Evaluation Index System of High-Quality Development

In terms of multi-dimensional nature, integrity, and coordination, this study follows the principles of science and comparability and constructs a comprehensive evaluation index system of high-quality development in the 3D system [26,27,28]. Table 1 reports different indicators of each system.

The high-quality development of economic systems includes upgrades to economic structure, technological innovation, and improvements to total factor productivity [29,30]. Social systems involve the coordinated and shared development of urban and rural areas, improved infrastructure, education and culture, social harmony, and stability [31,32,33]. The indicators of ecological environment system development originate from the 2015 Technical Criterion for Ecosystem Status Evaluation published by the Ministry of Ecology and Environment [34], covering vegetation coverage, water density, pollution load, and biological abundance.

**Table 1 ijerph-20-04305-t001:** Comprehensive evaluation index system of high-quality development in the three-dimensional system during 2010–2019.

System	Dimension	Criterion Layer	Indicator	Attributes	Weight	Source
Economy	Economic structure upgrading	Economic structure	Industrial structure rationalization index	−	0.0303	Wang et al., 2022 [31]
			Industrial structure upgrading index	+	0.0430	Wang et al., 2022 [31]
		Investment structure	Number of start-ups	+	0.0451	Wang et al., 2022 [31]
			Ratio of investment in the tertiary industry	+	0.0315	Wang et al., 2022 [31]
			Number of VCPE investments	+	0.0509	Wang et al., 2022 [31]
		Consumption structure	Consumer Price Index	−	0.0203	Wang et al., 2022 [31]
			Ratio of local government budgetary expenditure	−	0.0081	Wang et al., 2022 [31]
		Trade structure	Number of new foreign projects for the year	+	0.0269	Wang et al., 2022 [31]
			Foreign investment in actual use/GDP	+	0.1043	Song et al., 2022 [27]
			Total import and export of goods/GDP	+	0.1169	Song et al., 2022 [27]
	Innovation-driven	Innovation input	R&D expenditures of industrial enterprises above designated size/Gross industrial output value	+	0.0629	Song et al., 2022 [27]
			R&D employees of industrial enterprises above designated size/employees of secondary industry	+	0.0943	Song et al., 2022 [27]
		Innovation output	Number of invention patent grants	+	0.0245	Wang et al., 2022 [31]
			Number of published utility model patents	+	0.0546	Wang et al., 2022 [31]
			Number of published design patents	+	0.0148	Wang et al., 2022 [31]
			New product sales revenue of large and medium-sized industrial enterprises/Gross industrial output value	+	0.0814	Wang et al., 2022 [31]
	Efficient resource allocation	Capital productivity	GDP/Total investment in fixed assets	+	0.0364	Wang et al., 2022 [31]
		Labor productivity	GDP/All employees	+	0.0632	Wang et al., 2022 [31]
		Energy productivity	GDP/million tce	+	0.0647	Fang et al., 2020 [33]
		Land productivity	Total grain production/Total arable area	+	0.0260	Wang et al., 2022 [31]
Society	Coordinated and shared development	Regional revenue sharing	Cities’ GDP per capita/Guangdong’s GDP per capita	+	0.0636	Wang et al., 2022 [31]
		Regional consumption sharing	Cities’ household consumption level/Guangdong’s average household consumption level	+	0.0442	Wang et al., 2022 [31]
		Urban and rural income coordination	Urban-rural income ratio	−	0.0043	Song et al., 2022 [27]
		Urban and rural consumption coordination	Urban-rural consumption ratio	−	0.0072	Song et al., 2022 [27]
	Improved infrastructure	Public transport	Number of buses for every 10,000 people	+	0.1659	Wong et al., 2013 [32]
		Transportation facility	Per capita urban road area	+	0.1331	Song et al., 2022 [27]
		Medical facility	Number of beds in hospitals and health centers (pieces) per capita	+	0.0139	Song et al., 2022 [27]
		Basic old-age facility	Ratio of basic endowment insurance	+	0.0224	Wong et al., 2013 [32]
	Education and culture	Education facility	Number of colleges and universities	+	0.2102	Fang et al., 2020 [33]
		Cultural facility	Public library holdings per capita	+	0.1248	Song et al., 2022 [27]
		Internet penetration rate	Capita Internet broadband access port	+	0.0656	Fang et al., 2020 [33]
		Teacher-student ratio of elementary education	Teacher-student ratio of secondary schools	+	0.0196	Wong et al., 2013 [32]
	Social harmony and stability	The emphasis of the service industry	Value-added of tertiary industry/GDP	+	0.0174	Wang et al., 2022 [31]
		Advance of service product	Density of starred hotels (five-star and four-star)	+	0.0661	Wang et al., 2022 [31]
		Optimization of service consumption	Total revenue of tourism/number of tourists	+	0.0314	Wang et al., 2022 [31]
		Stability of social employment	Cities’ registered urban unemployed population	−	0.0044	Wang et al., 2022 [31]
		Safety production accident rate	Cities’ death rate of production safety accidents with a GDP of RMB 100 million	−	0.0060	Wang et al., 2022 [31]
Ecological environment	Biological abundance	Biological abundance index	Biodiversity index, ecological environment quality index (forestry, grassland, water wetland, cropland, construction land, unused land)	/	0.35	Ministry of Ecology and Environment, 2015 [34]
	Vegetation coverage	Vegetation coverage index	Normalized Difference Vegetation Index (NDVI)	/	0.25	Ministry of Ecology and Environment, 2015 [34]
	Water density	Water density index	River length, water area, water resources	/	0.15	Ministry of Ecology and Environment, 2015 [34]
	Land stress	Land stress index	Severe erosion intensity, moderate erosion intensity, construction land, other land stress	/	0.15	Ministry of Ecology and Environment, 2015 [34]
	Pollution load	Pollution load index	SO_2_ emission, ammonia-nitrogen emission, dust emission, amount of solid waste discarded, chemical oxygen demand	/	0.10	Ministry of Ecology and Environment, 2015 [34]
	Environmental constraints	Environmental constraints index	Emergent environmental incidents, environmental pollution	/	Obligatory	Ministry of Ecology and Environment, 2015 [34]

Notes: “+” and “−” represent the positive and negative indexes, respectively. A higher positive index indicates a more remarkable improvement. Source: arranged by the authors.

### 2.3. The Entropy Weight TOPSIS Method

The entropy weight TOPSIS method is an objective approach that reveals the utility of each indicator and avoids the disturbance of subjective elements [35]. Firstly, the extremum method is selected as the data standardization method. Then, the entropy weight method is applied to calculate the weight of each indicator. Lastly, the TOPSIS method is used to measure and rank the index of high-quality development. The entropy weight TOPSIS approach is suitable for evaluating emerging new objects [36]. This paper calculates the level of high-quality development in Guangdong province based on the entropy weight TOPSIS method, as follows:

#### 2.3.1. Standardization and Processing

Dimensionless processing is performed on the matrix to eliminate the influence of the index dimension. The extremum method is a data standardization method used to calculate economic and social high-quality development in Guangdong province:(1)$$\mathrm{Positive}\;\mathrm{indicator}:\;Z_{ij}=\frac{\left(X_{ij}-X_{min}\right)}{\left(X_{max}-X_{min}\right)}$$
(2)$$\mathrm{Negative}\;\mathrm{indicator}:\;Z_{ij}=\frac{\left(X_{max}-X_{ij}\right)}{\left(X_{max}-X_{min}\right)}$$
where $Z_{ij}$ is the value after standardization; *i* and *j* represent a city and an indicator, respectively; $X_{max}$ and $X_{min}$ are the maximum and minimum of $X_{ij}$; $X_{ij}$ is the value before standardization with a positive shift (0.0001); and $Z_{ij}\in \left[0.0001,\;1.0001\right]$.

#### 2.3.2. Calculation of Index Weight

The entropy weight method is applied to define each indicator’s weight and avoid the impact of human interference on the indicators:(3)$$\mathrm{Normalization}:\;P_{ij}=\frac{Z_{ij}}{\sum_{i=1}^{n}Z_{ij}}$$

$P_{ij}$ is the normalized value of $Z_{ij}$.
(4)$$\mathrm{Information}\;\mathrm{entropy}\;\mathrm{value}:\;E_{j}=-k\sum_{i=1}^{n}P_{ij}\ln P_{ij}$$
where k=1/lnn; $E_{j}$ is the entropy value of indicator *j*.
(5)$$\mathrm{Weight}:\;W_{j}=\frac{1-E_{j}}{\sum_{j=1}^{m}D_{j}}$$

$W_{j}$ is the weight of various indicators in Table 1.

#### 2.3.3. TOPSIS Method

The positive and negative ideal solutions are defined according to the weighted matrix of the high-quality development index. We also calculated the Euclidean distance between the solutions, the closeness of the evaluated target object, and the optimal solution.
(6)$$\mathrm{The}\;\mathrm{weighted}\;\mathrm{matrix}:\;R={\left(r_{ij}\right)}_{n\times m}$$
where $r_{ij}=\left(W_{j}\times Z_{ij}\right)$.

The positive and negative ideal solutions are $Q_{j}^{+}$ and $Q_{j}^{-}$:
(7)$$Q_{j}^{+}=\left(Q_{1}^{+},Q_{2}^{+},\ldots \;Q_{m}^{+}\right)$$
(8)$$Q_{j}^{-}=\left(Q_{1}^{-},Q_{2}^{-},\ldots \;Q_{m}^{-}\right)$$

The Euclidean distance between the positive and negative ideal solutions is calculated as $d_{i}^{+}$ and $d_{i}^{-}$:(9)$$d_{i}^{+}=\sqrt{\sum_{j=1}^{m}{\left(Q_{j}^{+}-r_{ij}\right)}^{2}}$$
(10)$$d_{i}^{-}=\sqrt{\sum_{j=1}^{m}{\left(Q_{j}^{-}-r_{ij}\right)}^{2}}$$

*C_i_* is the closeness of the evaluated target object and the optimal solution:(11)$$C_{i}=\frac{d_{i}^{-}}{d_{i}^{+}+d_{i}^{-}}$$

*C_i_* is between 0 and 1, representing the high-quality development level of city *i* in various systems.

### 2.4. Coupling Coordination Degree Model

The high-quality development index of the economic and social systems of Guangdong province is calculated with the entropy weight TOPSIS approach. Furthermore, this paper utilized the coupling coordination degree model to analyze the interactions among economic, social, and ecological environment systems with a high-quality development index.

#### 2.4.1. Types of Coupling Degree Models

Coupling is a concept encountered in physical electronics [37] that can help in analyzing the interactions of two or more systems [38,39]. This study applies the coupling degree model to analyze the reciprocity of the 3D system of economy, society, and ecological environments. A higher coupling value implies a more positive effect on high-quality development and vice versa. Table 2 summarizes four types of coupling degree models:(12)$$C=3{\left\{\frac{\left(U_{1}\times U_{2}\times U_{3}\right)}{{\left(U_{1}+U_{2}+U_{3}\right)}^{3}}\right\}}^{1/3}$$
where *C* is the coupling degree of the high-quality development of the 3D system of economy, society, and ecological environments; C∈[0, 1]; and $U_{1}$, $U_{2}$, and $U_{3}$ indicate the high-quality development indexes of the economic, social, and ecological environments systems, respectively.

#### 2.4.2. Types of Coupling Coordination Degree Models

The coupling coordination degree model is widely used in the study of the interactions and coordinated development between systems, reflecting various states of coordination between the systems [41]. The formula is as follows:(13)$$D=\sqrt{C\times T}$$
(14)$$T=aU_{1}+bU_{2}+cU_{3}$$
where $U_{1}$, $U_{2}$, and $U_{3}$ represent the high-quality development indexes of the economic, social, and ecological environment systems, respectively; *D* represents the coupling coordinated degree of the three systems; *T* represents the coupling degree of three systems; and *a*, *b*, and *c* are undetermined coefficients, defined as the contribution of each system to national development. Referring to Fan et al. (2019) [42] and Lv et al. (2019) [43], we consider all three coefficients to be equally important, and they are assumed to be 0.3333.
(15)$$T_{1}=aU_{1}+bU_{2}$$
(16)$$T_{2}=aU_{1}+cU_{3}$$
(17)$$T_{3}=bU_{2}+cU_{3}$$

Table 3 shows five levels of coupling coordination degree for high-quality development [44,45].

### 2.5. Data Sources

The data used in this study mainly come from the Guangdong Statistical Yearbook, the China City Statistical Yearbook, and the Statistical Yearbook of various municipalities for 2010–2019. The ecological index is derived from the Department of Ecology and Environment of Guangdong Province. Given the different criteria and missing values, this study collects data from Guangdong’s science and technology statistics data, environment bulletin, and other data platforms, such as the Guangzhou Municipal Government Data Platform. The interpolation method is used to fill in any missing data. Figure 2 illustrates the flowchart of methods in this study.

## 3. Results

### 3.1. Measurement of High-Quality Development of Economic, Social, and Ecologic Environment Systems

As illustrated in Table 4, the economic system’s high-quality development score averaged between 0.32 and 0.35. Nevertheless, there was considerable heterogeneity in the high-quality development level of the economic system between cities. Shenzhen and Zhuhai had the highest mean of 0.61, whereas Jieyang, Maoming, and Yangjiang had the lowest mean. Regionally, the Pearl River Delta was a mid-high-level region with a maximum value of 0.48. Eastern Guangdong was a middle-level region with a mean of 0.25.

Regarding the social system, Table 5 shows that Guangdong’s high-quality development scores averaged between 0.13 and 0.16, indicating a relatively slow rise in high-quality development during 2010–2019. The uneven distribution of social resources centered on developed cities resulted in a considerable regional income gap and a significant infrastructure disparity. However, there is a massive divergence within the province concerning the social system. Guangzhou and Shenzhen had the highest score of over 0.50, whereas Jiangmen, Shantou, Zhanjiang, and Shaoguan had the lowest scores of less than 0.10. It is supported by unbalanced regional and urban-rural development, leading to more public service resources in Guangzhou, Shenzhen, and other cities in Pearl River Delta, such as higher education, medical resources, and libraries. Table 5 also reveals a slight increase in the value of high-quality development of the social system, except for the low score of less than 0.10. The high-quality development score of the social system in the Pearl River Delta averaged 0.26 during 2010–2019, implying a higher-quality development of the social system than in other regions.

As can be seen from Table 6, the high-quality development score of the ecological environment system averaged between 0.68 and 0.75, showing a steady rise during 2010–2019. Shaoguan, Qingyuan, and Heyuan ranked the highest, with scores of more than 0.80, whereas Guangzhou, Foshan, Dongguan, Zhongshan, Shantou, and Zhanjiang ranked the lowest, with scores of less than 0.67. These results suggest that hilly cities have a higher-quality development of the ecological environment compared with coastal cities. In addition, Northern Guangdong had the highest score for the ecological environment index, reaching 0.79. In contrast, the scores of Pearl River Delta, Eastern, and Western Guangdong ranged from 0.69 to 0.72. These data demonstrate that Northern Guangdong is in a better ecological environment development situation than other regions, with more wildlife habitats and green agricultural products.

### 3.2. Measurement of High-Quality Development in Guangdong Province

Table 7 shows the comprehensive development index of 21 cities in Guangdong during 2010–2019. Guangdong’s high-quality development experienced a slow increase, with the mean ranging from 0.31 to 0.40. The comprehensive development index was lower than 0.35 during 2010–2015 owing to the influx of the cheap labor force, which led to increased environmental awareness, severe air pollution, and appropriated land resources. Thus, the local ecosystem was confronted by a series of environmental issues, exerting an indirect and adverse impact on the high-quality development of Guangdong province. However, the comprehensive development index has increased since 2015, which is attributed to enhanced scientific and technological innovation ability, enterprise transformation, and environmental regulation. All these actions led to economic restructuring, increased environmental awareness, and improved new infrastructure. Nonetheless, Shenzhen, Guangzhou, Zhuhai, and Dongguan ranked in the top four cities in terms of the comprehensive development index, with scores higher than 0.56. In contrast, Chaozhou, Jieyang, Zhanjiang, Maoming, and Yunfu ranked relatively low, with scores lower than 0.19.

During 2010–2019, the high-quality development of nine cities in the Pearl River Delta was more convergent. The first-echelon cities comprised Shenzhen, Guangzhou, Zhuhai, and Dongguan; the second-echelon cities comprised Foshan and Huizhou in 2010 and then also Jiangmen in 2018; and the third-echelon cities comprised Zhaoqing and Zhongshan. It should be noted that there needs to be more high-quality development in cities. In 2010, the high-quality development scores of Guangzhou, Zhuhai, and Dongguan were 0.61, 0.58, and 0.52, rising slowly to 0.71, 0.65, and 0.57 in 2019, respectively. Shenzhen showed a downward trend from 0.85 to 0.78 during 2010–2019 and was confronted with growing environmental stress due to economic slowdown, industrial structure upgrading, and labor inflow. The high-quality development of Foshan and Huizhou exhibited steady growth from 2010–2019, reaching 0.46 and 0.49 in 2019, respectively. Jiangmen experienced rapid growth in high-quality development from 0.31 in 2010 to 0.43 in 2019. It is worth mentioning that the high-quality development score of Zhongshan was more than 0.40 in most years but experienced a lower-volatility growth because the city was under tremendous pressure from industrial structure upgrading, social development, and environmental protection. Zhaoqing had a better ecological environment, but its economic and social development showed slow growth scaling from 0.27 to 0.37.

The spatial patterns of high-quality development in 2010, 2015, and 2019 are depicted in Figure 3. There was a vast difference in high-quality development among the cities over five different periods. It is generally agreed that regional coordination development is under tremendous pressure. Shenzhen, Guangzhou, Zhuhai, and Dongguan were high-level cities, implying that estuary cities succeeded in economic transition, technological innovation, and industrial agglomeration, leading to higher-quality development of the economy. More importantly, better infrastructure, high-quality workers, social stability, and higher environmental standards contribute to social harmony and environmental restoration. Foshan and Huizhou were high-level cities in 2010, as was Jiangmen in 2019. The mid-level cities were Zhongshan, Zhaoqing, Qingyuan, Shaoguan, Heyuan, and Meizhou in 2019, whereas the others were mid-low-level cities. Zhanjiang was a low-level city due to a higher proportion of the population being in Northern Guangdong, more significant pressure from economic transition, and poorer public facilities.

Guangzhou, Shenzhen, Zhuhai, and Dongguan were the core cities of high-quality development in Guangdong province. Guangdong’s high-quality development index decreased from the Pearl River Delta estuary cities to its edge. The high-quality development of Guangdong shows a spatial distribution pattern of high values in the Pearl River Delta and low values in other regions. Northern, Eastern, and Western Guangdong should adjust their development strategies to promote high-quality regional development, including the continuation of the industry transfer from the Pearl River Delta and rural revitalization. As the core region of Guangdong, the Pearl River Delta is divided into three metropolitan areas. Shenzhen, Dongguan, and Huizhou are eastern megalopolis; Guangzhou, Foshan, and Zhaoqing are central megalopolis; and Zhuhai, Zhongshan and Jiangmen are western megalopolis. High-quality economic development depends on technology innovation and factor productivity. Higher-quality labor force, infrastructure, and public services (i.e., education and health care) contribute to social harmony and stability. In addition, environmental protection and restoration receive more attention while promoting economic and social development.

### 3.3. Coupling Degree Analysis of High-Quality Development of Three Dimensions

Table 8 shows that the coupling degree of three dimensions demonstrated a slow growth, ranging from 0.55 to 1.00. In 2010, Guangzhou, Shenzhen, Foshan, and Dongguan were the top four in terms of the coupling degrees of the three dimensions. These cities were in a period of coordinated coupling, with a value of over 0.90. On the other hand, Zhuhai, Zhongshan, Huizhou, Shantou, Jiangmen, and Zhaoqing were in a period of benign coupling. Additionally, Shaoguan, Heyuan, Meizhou, Shanwei, Yangjiang, Zhanjiang, Maoming, Jieyang, Yunfu, Chaozhou, and Qingyuan were in a period of low coupling, with a value of less than 0.70. In 2019, Zhuhai became the city with the higher coupling degree among the three dimensions, whereas the cities with low coupling degrees were Heyuan, Meizhou, Shanwei, Jieyang, and Yunfu. More importantly, there were 11 cities with benign coupling, implying a mutual promotion among the economic, social, and ecological environment systems.

Figure 4 provides an overview of the spatial distribution of the three dimensions’ coupling degree in 21 cities in 2010, 2015, and 2019. In 2010, Guangzhou, Shenzhen, Foshan, and Dongguan were inner-circle cities of coupling degree; Zhongshan, Zhuhai, Jiangmen, Huizhou, and Zhaoqing were middle-circle cities. In Eastern Guangdong, Shantou was a middle-circle city, whereas others were outer-circle cities. In 2015, Zhuhai was an inner-circle city, and peripheral cities of the Pearl River Delta mainly stayed in the middle circle. Zhanjiang and Shaoguan were central cities of Western and Northern Guangdong, respectively. In 2019, the estuary cities of the Pearl River Delta still remained in the inner circle. However, more cities in the period of benign coupling expanded the scope of the middle circle and narrowed the range of the outer circle. Spatially, the coupling degree took Guangzhou, Shenzhen, Foshan, Dongguan, and Zhuhai as the core, other cities in Pearl River Delta, Shantou, Zhanjiang, and Shaoguan as the sub-core, and still other cities as the edge.

### 3.4. Coupling Coordination Analysis of High-Quality Development of Three Dimensions

Table 9 presents the classification of the coupling coordination degree of the three studied dimensions in 21 cities in Guangdong based on Formulas (13) and (14), which ranged from 0.43 to 0.81 during 2010–2019. The coupling coordination degree of most cities gradually increased, with means of 0.46 and 0.80, which represented the basic coordination and moderate coordination stages, whereas Shenzhen first decreased and then increased. Guangzhou, Shenzhen, Foshan, Dongguan, Zhuhai, and Zhongshan were in moderate coordination in 2010, and Huizhou and Jiangmen joined them in 2019. Guangzhou, Shenzhen, and Zhuhai were the pillar cities in coupling coordination of high-quality regional development. Nonetheless, the number of cities in basic coordination showed a downward trend, declining from 15 to 13 during 2010–2019. Additionally, the most remarkable result to emerge from the data is that the coupling coordination degree of each city grew slowly, implying higher-quality development during the study period.

As shown in Figure 5, the Pearl River Delta had a higher degree of coupling coordination of the three studied dimensions in 2010, 2015, and 2019. The increase in the number of cities with a higher degree of coupling coordination demonstrates that it has witnessed rapid growth. Meanwhile, the coupling coordination of 12 cities in Eastern, Western, and Northern Guangdong has remained the same for over ten years. These cities were in the basic coordination stage, but there were significant gaps in high-quality coordination between the Pearl River Delta and other regions in Guangdong. Spatially, the coupling coordination of the three dimensions took the Pearl River Delta (except Zhaoqing) as the core and others as the edge.

## 4. Discussion

High-quality development is conducive to increasing living standards, addressing issues of social justice, and achieving sustainable development. It involves the coordination development of economic, social, and ecological environments. Nonetheless, existing studies pay attention to the relationship between the two systems. This study established a high-quality development index system covering economic, social, and ecological systems. Then this study applied the TOPSIS method to calculate the level of high-quality development in Guangdong province and used the coupling coordination model to analyze the interactions among economic, social, and ecological systems from 2010–2019. Our results suggest that there was a slow increase in Guangdong’s high-quality development from 2010–2019. The Pearl River Delta had the highest level of high-quality development, whereas Eastern Guangdong had the lowest level. There was a significant divergence in the level of high-quality development at the city level. The coupling degree of high-quality development in Guangdong grew slowly, with more cities entering the stage of benign coupling. The coupling coordination degree of high-quality development in Guangdong remained stable, and the Pearl River Delta had a higher degree of coupling coordination degree of the three studied systems.

The results demonstrate that Guangzhou, Shenzhen, Foshan, Dongguan, and Zhuhai have higher-quality development of economic and social systems than other cities. In comparison, Northern Guangdong has higher-quality development of the ecological system. The ecological space is mainly distributed in hilly areas and mountains in Northern Guangdong, which is an important area for protecting and restoring key ecosystems in Guangdong [46]. Previous literature argued that the development of the economy is advancing in the direction of high ecological carrying capacity [47,48]. These findings are consistent with our analysis in that frequent human activities make ecosystem restoration more difficult. The highest level of high-quality development in the Pearl River Delta receives benefits from preferential policies, the rapid growth of high-tech industries, and improved infrastructure facilities [49,50]. More importantly, the divergence of high-quality development among cities in Guangdong implies an imbalance of regional development concerning the economy, society, and ecology in the Guangdong province.

Moreover, the data of coupling degree indicates that Guangdong takes Guangzhou, Shenzhen, Foshan, Dongguan, and Zhuhai as the core, and Shantou, Zhanjiang, Shaoguan, and other cities in the Pearl River Delta as the sub-core, as well as other cities in Guangdong as the edge. Temporally, the coupling coordination of each city shows an upward trend, implying that Guangdong is inclined to enhance the high-quality regional development of the three dimensions. Wang et al. supports the notion that Guangdong takes the lead in entering the high-coupling stage [51]. Our data demonstrates that economic growth and innovation-driven economy in estuary cities of the Pearl River Delta contribute to the spatial pattern described above. Population agglomeration, improving infrastructure, and strict environmental regulations result in mutual promotion and better coupling among the three dimensions in Guangdong province. Shantou, Zhanjiang, and Shaoguan take advantage of a complete industry foundation and enormous economic potential, setting a solid foundation for promoting high-quality development.

The coupling coordination analysis of high-quality development suggests that Guangdong, especially the Pearl River Delta, experienced a slight growth in high-quality development. The findings match those observed in previous studies, but they discovered the coupling coordination between the economy and eco-environment in the Pearl River Delta [52]. Guangzhou, Shenzhen, and Zhuhai play a crucial role in leading high-quality provincial development due to a reasonable economic structure and attach great significance to technology, talents, and capital [53,54]. As a result, local governments and people realize the significance of environmental protection and take a series of measures to restore the ecosystem. In the long run, these actions contribute to building an economic structure conducive to green, low-carbon, and circular development. These results are well in line with previous studies concerning the coupling coordination between two systems [47,51], whereas our study further investigates the interactions among three systems.

The sensitivity analysis performed in this section validates the robustness of the entropy method [55,56,57,58]. The variation coefficient is selected to test whether the different values of the index weight led to a substantial change in the evaluation results [59,60]. According to the results, we found that the sensitivity fluctuation of the results obtained by this method was quite small and relatively stable. Therefore, the evaluation results are objective and reliable.

## 5. Conclusions

As the most economically developed province in China, Guangdong province is expected to play a significant role in boosting high-quality regional development. In this study, the entropy weight TOPSIS model was used to assess the high-quality development of the 21 cities in Guangdong. The coupling coordination pattern of the three-dimensional system was analyzed using the coupling coordination degree model.

The main findings of the paper are as follows. First, the high-quality development of most cities shows slow growth. Guangzhou and Shenzhen have higher-quality development of economic and social systems than other cities, whereas Northern Guangdong has higher-quality development of the ecological environment, which can be attributed to rapid economic growth in the Pearl River Delta and less anthropogenic activities in Northern Guangdong. Second, the coupling degree of high-quality development of the three-dimensional system in Guangdong Province mainly presents three periods: low coupling, benign coupling, and coordinated coupling; and the degree of coupling coordination is in two stages: basic and moderate. Half of the cities in Guangdong province entered the stage of benign coupling during 2010–2019. Finally, except Zhaoqing, all cities in the Pearl River Delta have a high coupling coordination degree of high-quality development of the three-dimensional system. In contrast, the scores of other cities in Guangdong are low. During the study period, Guangdong transformed from a low coupling period to a coordinated coupling period, indicating Guangdong’s high-quality development entered a higher level of the regional development stage.

This study puts forward some suggestions for Guangdong’s high-quality development and coordination development. Firstly, it is necessary to differentiate economic development strategies and reinforce technological innovations at the city level for regional economic structure improvement. Our results indicate a low degree of high-quality development of economic and social systems in Northern and Eastern Guangdong. In response to the regional development pattern of “one core, one belt, and one zone”, a differentiated economic transformation strategy should be implemented to optimize industrial structures. We discovered that Northern Guangdong has a higher-quality development in terms of ecological environment compared with other regions. Therefore, cities should place emphasis on clean production technology by increasing the funds for scientific innovation, building up a diversified financing support system, and creating an environment of the rational and orderly flow of innovation elements.

Secondly, a stable and harmonious society could be maintained through a better social security network, equal access to public service, and higher-quality employment. Guangdong’s urbanization rate has experienced rapid growth with the growing economy. Thus, a basic public service system should be established to serve population mobility. In addition, compared with other regions, Northern Guangdong has a lower coupling coordination degree of high-quality development, which can be attributed to the poor access to public service and low educational level. Thus, improving infrastructures brings about high-quality employment, helping local people get rid of poverty.

Finally, distinguished environmental regulations and tools aid in the cooperative governance of the local environment. Various environmental regulation tools should be comprehensively applied, including pollution discharge fees, environmental subsidies, and emission reduction control. Enterprises are encouraged to carry out green production, reduce the production cost of green transformation, and get more ecological, social, and environmental benefits.

There are still several limitations to this study. Firstly, the evaluation system should consider more soft power indexes, such as enterprise vitality, resident satisfaction, and market mechanisms. However, it is not easy to quantify these subjective indicators. It is hoped that more subjective indicators will be incorporated into the high-quality development system. Secondly, the data of this study mainly reflects the urban area, ignoring rural development. It may not be sufficient to fully reflect the actual high-quality development of Guangdong province. Future research should attach great importance to urban and rural coordination development with more rural data, considering the high-quality development of rural areas. Lastly, the classic TOPSIS technique is insufficient to cope with subjectivity and incomplete information. If we consider more subjective indicators in the future, the TOPSIS method should be integrated with other theories, such as the fuzzy set theory.

## Figures and Tables

**Figure 1 ijerph-20-04305-f001:**
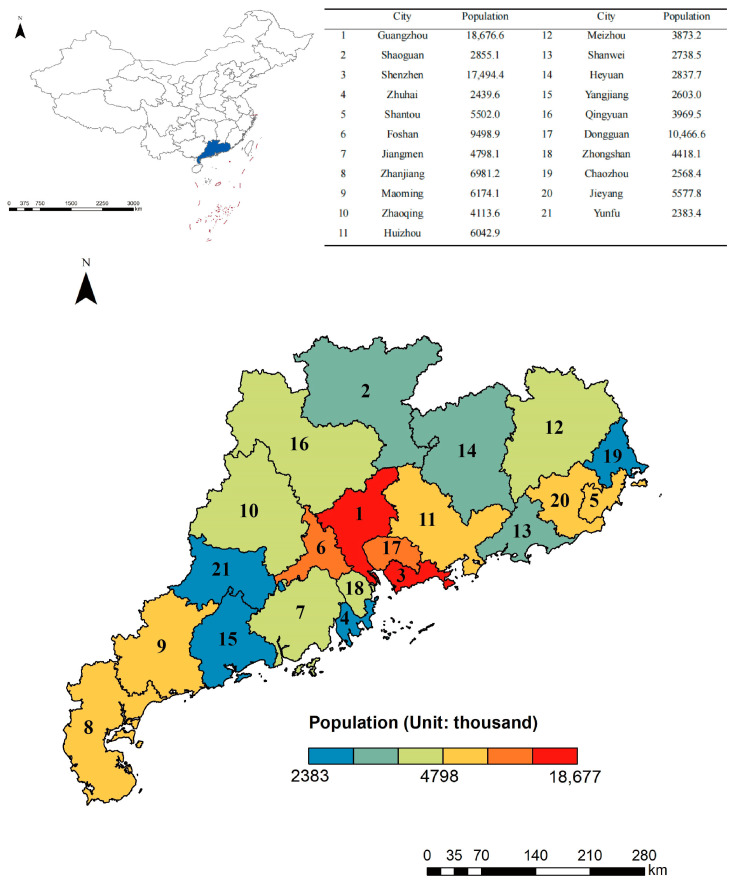
Administrative divisions and population distribution in the Guangdong province of China. The dark blue area in the upper left picture represents the Guangdong province in China. The unit of the population in the upper right table is a thousand people. The population data originates from the seventh national census.

**Figure 2 ijerph-20-04305-f002:**
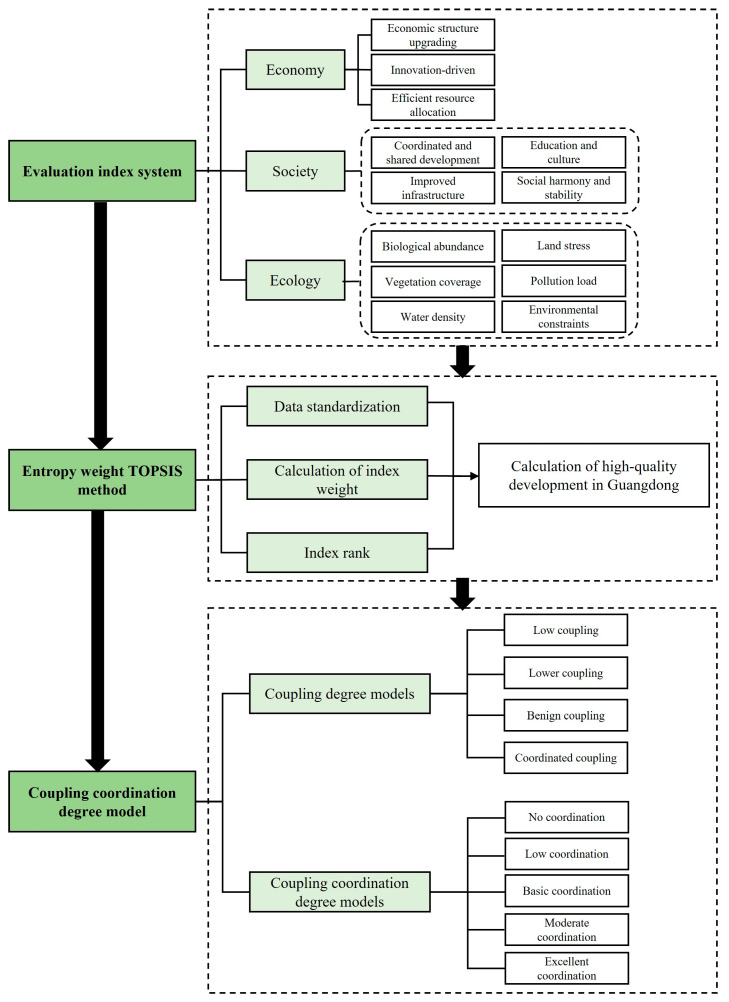
Flowchart of methodology.

**Figure 3 ijerph-20-04305-f003:**
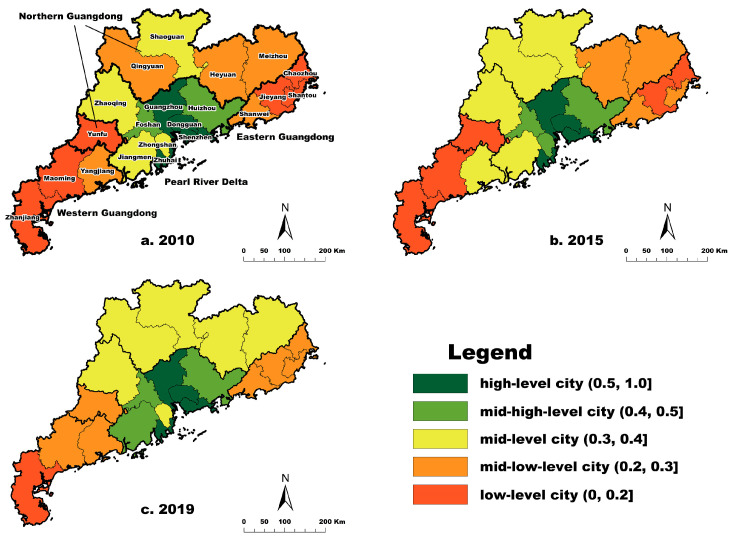
Spatial distribution of high-quality development in Guangdong.

**Figure 4 ijerph-20-04305-f004:**
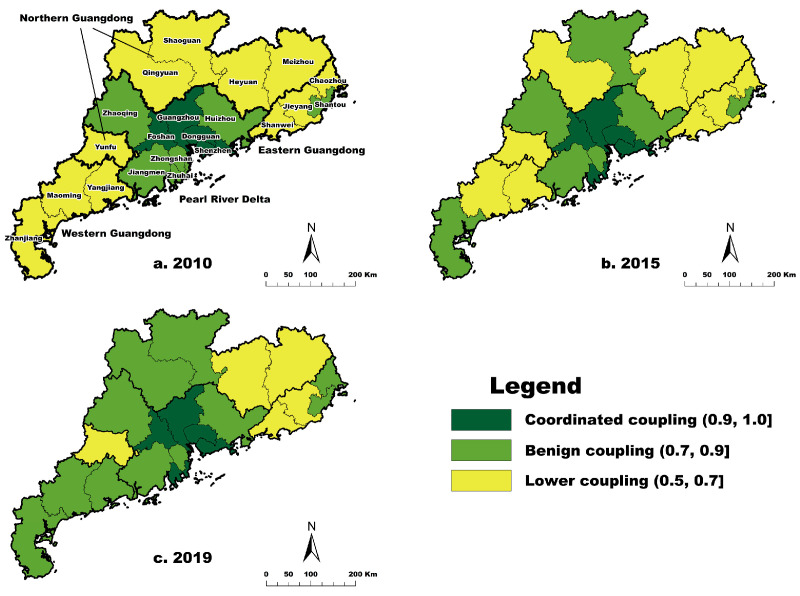
Spatial distribution of coupling degree of high-quality development of three dimensions in Guangdong.

**Figure 5 ijerph-20-04305-f005:**
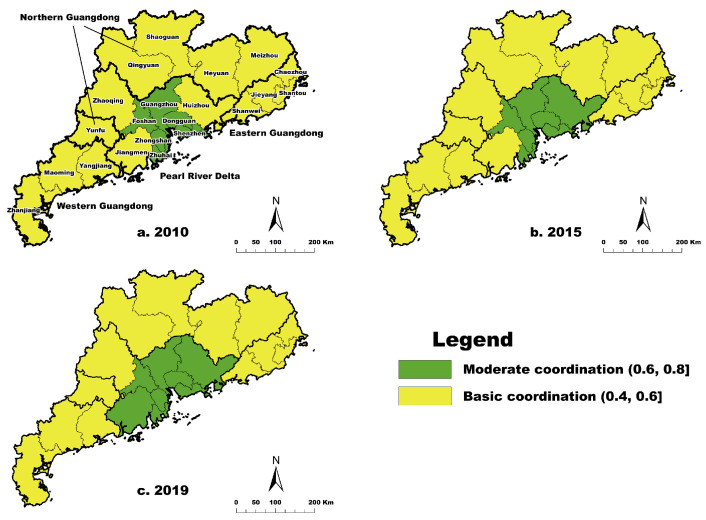
Spatial distribution of coupling coordination degree of high-quality development of three dimensions in Guangdong.

**Table 2 ijerph-20-04305-t002:** The classification of coupling degree models.

Coupling (C)	Category	Characteristics
0.00 ≤ C ≤ 0.50	low coupling	There is a game between the economy, society, and ecological environments, but the interactions with each other are weak. When C = 0, there is no interaction among the three dimensions.
0.50 < C ≤ 0.70	lower coupling	There is a strong interaction between the economy, society, and ecological environments. The system with higher-quality development dominates the development, whereas other systems show a downward trend.
0.70 < C ≤ 0.90	benign coupling	There is a better balance between each dimension, and the coupling of the three dimensions becomes benign.
0.90 < C ≤ 1.00	coordinated coupling	There is a stronger coupling among the three dimensions. When C = 1.0, the two systems have a higher coupling coordination degree of high-quality development.

Source: based on the study of Wang & Tang (2018) [40].

**Table 3 ijerph-20-04305-t003:** The classification of coupling coordination degree.

Coupling Coordination Degree (D)	Stage	Category	Characteristics
0.00 ≤ D ≤ 0.20	No coordination	Low-level city	Only one system develops better, but the others are restricted. There is no coordination.
0.20 < D ≤ 0.40	Low coordination	Lower-level city	Only one system takes the leading role, and the others develop slowly. Thus, a series of issues arise.
0.40 < D ≤ 0.60	Basic coordination	Average-level city	Only one system develops slowly and is more concerned about high-quality development.
0.60 < D ≤ 0.80	Moderate coordination	Higher-level city	The city makes excellent progress in coping with issues of high-quality development and improving the coordination of the three dimensions.
0.80 < D ≤ 1.00	Excellent coordination	High-level city	There is excellent coordination among the three dimensions. As a result, various human needs could be satisfied.

Source: based on the study of Dong et al. (2021) [44] and Fang et al. (2021) [45].

**Table 4 ijerph-20-04305-t004:** High-quality development score of the economic system during 2010–2019.

Region	City	2010	2011	2012	2013	2014	2015	2016	2017	2018	2019	Mean
Pearl River Delta	Guangzhou	0.44	0.44	0.45	0.46	0.46	0.47	0.48	0.51	0.50	0.50	0.47
	Shenzhen	0.63	0.62	0.64	0.62	0.61	0.60	0.61	0.61	0.60	0.60	0.61
	Foshan	0.41	0.41	0.43	0.43	0.44	0.42	0.42	0.45	0.43	0.43	0.43
	Dongguan	0.51	0.48	0.49	0.50	0.51	0.52	0.52	0.49	0.51	0.51	0.51
	Zhuhai	0.60	0.61	0.61	0.61	0.62	0.60	0.60	0.62	0.61	0.61	0.61
	Huizhou	0.49	0.48	0.51	0.51	0.49	0.45	0.46	0.48	0.50	0.49	0.49
	Zhaoqing	0.40	0.37	0.37	0.37	0.36	0.36	0.29	0.29	0.31	0.29	0.34
	Jiangmen	0.38	0.34	0.37	0.35	0.35	0.36	0.34	0.38	0.42	0.44	0.37
	Zhongshan	0.40	0.41	0.42	0.42	0.42	0.41	0.41	0.46	0.44	0.42	0.42
	Mean	0.47	0.46	0.48	0.47	0.47	0.47	0.46	0.48	0.48	0.48	-
Eastern Guangdong	Shantou	0.27	0.27	0.26	0.23	0.25	0.26	0.26	0.29	0.29	0.29	0.27
	Chaozhou	0.24	0.22	0.22	0.21	0.21	0.17	0.20	0.20	0.21	0.19	0.21
	Jieyang	0.18	0.18	0.16	0.16	0.17	0.17	0.19	0.19	0.23	0.22	0.19
	Shanwei	0.32	0.32	0.30	0.26	0.24	0.25	0.23	0.26	0.27	0.25	0.27
	Mean	0.25	0.25	0.24	0.22	0.22	0.21	0.22	0.24	0.25	0.24	-
Western Guangdong	Zhanjiang	0.20	0.19	0.17	0.18	0.21	0.19	0.19	0.22	0.24	0.24	0.20
	Maoming	0.19	0.18	0.18	0.17	0.19	0.18	0.17	0.20	0.21	0.20	0.19
	Yangjiang	0.21	0.20	0.17	0.17	0.16	0.16	0.17	0.17	0.19	0.18	0.18
	Mean	0.20	0.19	0.17	0.17	0.19	0.18	0.18	0.20	0.21	0.21	-
Northern Guangdong	Shaoguan	0.35	0.33	0.29	0.29	0.28	0.24	0.26	0.27	0.32	0.32	0.30
	Qingyuan	0.28	0.28	0.27	0.26	0.24	0.25	0.26	0.28	0.30	0.33	0.28
	Meizhou	0.22	0.20	0.21	0.20	0.23	0.24	0.24	0.26	0.26	0.24	0.23
	Heyuan	0.23	0.22	0.20	0.19	0.20	0.20	0.21	0.24	0.23	0.27	0.22
	Yunfu	0.21	0.20	0.20	0.22	0.20	0.19	0.17	0.18	0.20	0.21	0.20
	Mean	0.26	0.25	0.23	0.23	0.23	0.22	0.23	0.25	0.26	0.27	-
Mean		0.34	0.33	0.33	0.33	0.33	0.32	0.32	0.34	0.35	0.34	-

Source: Based on the authors’ calculation.

**Table 5 ijerph-20-04305-t005:** High-quality development score of the social system during 2010–2019.

Region	City	2010	2011	2012	2013	2014	2015	2016	2017	2018	2019	Mean
Pearl River Delta	Guangzhou	0.49	0.50	0.51	0.52	0.52	0.52	0.52	0.52	0.53	0.53	0.52
	Shenzhen	0.51	0.51	0.51	0.51	0.51	0.51	0.50	0.49	0.48	0.46	0.50
	Foshan	0.21	0.23	0.22	0.17	0.19	0.20	0.19	0.21	0.22	0.22	0.20
	Dongguan	0.31	0.34	0.35	0.33	0.37	0.39	0.39	0.36	0.36	0.37	0.36
	Zhuhai	0.23	0.24	0.24	0.27	0.31	0.33	0.30	0.30	0.30	0.30	0.28
	Huizhou	0.10	0.10	0.10	0.11	0.11	0.12	0.11	0.12	0.12	0.12	0.11
	Zhaoqing	0.08	0.08	0.08	0.09	0.09	0.09	0.08	0.11	0.13	0.13	0.10
	Jiangmen	0.08	0.08	0.09	0.08	0.08	0.10	0.10	0.11	0.11	0.10	0.09
	Zhongshan	0.15	0.15	0.15	0.15	0.16	0.16	0.15	0.15	0.15	0.16	0.15
	Mean	0.24	0.25	0.25	0.25	0.26	0.27	0.26	0.26	0.27	0.27	-
Eastern Guangdong	Shantou	0.07	0.07	0.07	0.07	0.07	0.08	0.08	0.09	0.09	0.09	0.08
	Chaozhou	0.06	0.06	0.07	0.07	0.06	0.07	0.07	0.07	0.07	0.11	0.07
	Jieyang	0.06	0.06	0.05	0.05	0.05	0.06	0.06	0.07	0.07	0.07	0.06
	Shanwei	0.07	0.05	0.05	0.04	0.04	0.06	0.07	0.07	0.07	0.06	0.06
	Mean	0.07	0.06	0.06	0.06	0.06	0.07	0.07	0.08	0.08	0.08	-
Western Guangdong	Zhanjiang	0.06	0.07	0.07	0.06	0.06	0.07	0.07	0.08	0.09	0.08	0.07
	Maoming	0.06	0.06	0.06	0.06	0.06	0.07	0.08	0.09	0.09	0.09	0.07
	Yangjiang	0.05	0.05	0.06	0.06	0.06	0.08	0.08	0.08	0.08	0.08	0.07
	Mean	0.06	0.06	0.06	0.06	0.06	0.07	0.08	0.08	0.09	0.08	-
Northern Guangdong	Shaoguan	0.06	0.06	0.06	0.06	0.07	0.08	0.09	0.09	0.10	0.08	0.07
	Qingyuan	0.05	0.05	0.05	0.05	0.06	0.07	0.07	0.08	0.08	0.07	0.06
	Meizhou	0.04	0.05	0.05	0.05	0.05	0.07	0.07	0.08	0.08	0.07	0.06
	Heyuan	0.04	0.04	0.04	0.04	0.05	0.06	0.07	0.08	0.08	0.07	0.06
	Yunfu	0.05	0.05	0.05	0.05	0.05	0.07	0.07	0.07	0.07	0.07	0.06
	Mean	0.05	0.05	0.05	0.05	0.06	0.07	0.07	0.08	0.08	0.07	-
Mean		0.13	0.14	0.14	0.14	0.14	0.15	0.15	0.16	0.16	0.16	-

Source: based on the authors’ calculation.

**Table 6 ijerph-20-04305-t006:** High-quality development score of the ecological environment system during 2010–2019.

Region	City	2010	2011	2012	2013	2014	2015	2016	2017	2018	2019	Mean
Pearl River Delta	Guangzhou	0.63	0.61	0.62	0.63	0.62	0.63	0.64	0.62	0.62	0.63	0.62
	Shenzhen	0.74	0.73	0.72	0.73	0.65	0.66	0.67	0.69	0.69	0.68	0.70
	Foshan	0.58	0.56	0.58	0.59	0.62	0.63	0.63	0.61	0.62	0.62	0.60
	Dongguan	0.61	0.58	0.60	0.61	0.60	0.60	0.62	0.60	0.60	0.61	0.60
	Zhuhai	0.75	0.72	0.73	0.73	0.70	0.69	0.69	0.71	0.71	0.71	0.72
	Huizhou	0.74	0.75	0.76	0.78	0.81	0.81	0.83	0.81	0.81	0.82	0.79
	Zhaoqing	0.73	0.72	0.74	0.74	0.80	0.80	0.80	0.82	0.83	0.82	0.78
	Jiangmen	0.72	0.69	0.72	0.72	0.75	0.74	0.75	0.77	0.77	0.77	0.74
	Zhongshan	0.68	0.65	0.67	0.68	0.67	0.66	0.67	0.64	0.63	0.63	0.66
	Mean	0.69	0.67	0.68	0.69	0.69	0.69	0.70	0.70	0.70	0.70	-
Eastern Guangdong	Shantou	0.66	0.66	0.67	0.68	0.66	0.66	0.68	0.67	0.68	0.68	0.67
	Chaozhou	0.68	0.66	0.67	0.69	0.70	0.71	0.74	0.77	0.76	0.77	0.71
	Jieyang	0.70	0.68	0.69	0.72	0.71	0.71	0.74	0.74	0.74	0.74	0.72
	Shanwei	0.67	0.65	0.67	0.67	0.73	0.72	0.74	0.82	0.83	0.82	0.73
	Mean	0.68	0.66	0.68	0.69	0.70	0.70	0.73	0.75	0.75	0.75	-
Western Guangdong	Zhanjiang	0.63	0.63	0.64	0.64	0.65	0.64	0.66	0.67	0.68	0.67	0.65
	Maoming	0.68	0.66	0.68	0.70	0.72	0.72	0.75	0.78	0.78	0.78	0.72
	Yangjiang	0.76	0.74	0.76	0.76	0.77	0.77	0.78	0.82	0.83	0.82	0.78
	Mean	0.69	0.68	0.69	0.70	0.71	0.71	0.73	0.76	0.76	0.76	-
Northern Guangdong	Shaoguan	0.78	0.77	0.79	0.79	0.82	0.83	0.84	0.85	0.85	0.85	0.82
	Qingyuan	0.76	0.74	0.76	0.78	0.81	0.82	0.83	0.84	0.85	0.85	0.80
	Meizhou	0.75	0.73	0.74	0.77	0.79	0.80	0.83	0.84	0.83	0.84	0.79
	Heyuan	0.78	0.76	0.77	0.78	0.80	0.81	0.83	0.83	0.83	0.84	0.80
	Yunfu	0.67	0.65	0.67	0.67	0.73	0.72	0.74	0.82	0.83	0.82	0.73
	Mean	0.75	0.73	0.75	0.76	0.79	0.80	0.81	0.84	0.84	0.84	-
Mean		0.70	0.68	0.70	0.71	0.72	0.72	0.74	0.75	0.75	0.75	-

Source: based on the authors’ calculation.

**Table 7 ijerph-20-04305-t007:** Comprehensive development index of 21 cities in Guangdong during 2010–2019.

Region	City	2010	2011	2012	2013	2014	2015	2016	2017	2018	2019	Mean
Pearl River Delta	Guangzhou	0.61	0.62	0.65	0.67	0.68	0.68	0.69	0.71	0.71	0.71	0.67
	Shenzhen	0.85	0.84	0.85	0.85	0.79	0.79	0.80	0.81	0.80	0.78	0.82
	Foshan	0.40	0.41	0.43	0.40	0.44	0.44	0.43	0.48	0.47	0.46	0.44
	Dongguan	0.52	0.50	0.53	0.54	0.57	0.60	0.60	0.56	0.57	0.57	0.56
	Zhuhai	0.58	0.58	0.60	0.62	0.65	0.66	0.63	0.66	0.65	0.65	0.63
	Huizhou	0.41	0.42	0.45	0.47	0.47	0.46	0.47	0.49	0.50	0.49	0.46
	Zhaoqing	0.30	0.27	0.29	0.30	0.33	0.34	0.32	0.35	0.37	0.35	0.32
	Jiangmen	0.31	0.27	0.32	0.32	0.33	0.35	0.34	0.39	0.42	0.43	0.35
	Zhongshan	0.37	0.37	0.40	0.41	0.41	0.40	0.40	0.43	0.42	0.39	0.40
	Mean	0.48	0.48	0.50	0.51	0.52	0.52	0.52	0.54	0.55	0.54	-
Eastern Guangdong	Shantou	0.19	0.20	0.20	0.19	0.19	0.21	0.22	0.25	0.26	0.26	0.22
	Chaozhou	0.16	0.13	0.15	0.17	0.18	0.17	0.20	0.24	0.24	0.24	0.19
	Jieyang	0.16	0.14	0.15	0.17	0.17	0.17	0.20	0.21	0.23	0.22	0.18
	Shanwei	0.27	0.25	0.26	0.27	0.25	0.26	0.26	0.29	0.29	0.28	0.27
	Mean	0.20	0.18	0.19	0.20	0.20	0.20	0.22	0.25	0.26	0.25	-
Western Guangdong	Zhanjiang	0.11	0.10	0.10	0.11	0.13	0.12	0.14	0.17	0.20	0.19	0.14
	Maoming	0.14	0.12	0.15	0.16	0.19	0.19	0.21	0.26	0.26	0.25	0.19
	Yangjiang	0.22	0.20	0.21	0.21	0.22	0.23	0.24	0.27	0.29	0.27	0.24
	Mean	0.16	0.14	0.15	0.16	0.18	0.18	0.20	0.23	0.25	0.24	-
Northern Guangdong	Shaoguan	0.32	0.29	0.30	0.30	0.32	0.31	0.33	0.34	0.38	0.37	0.32
	Qingyuan	0.24	0.23	0.26	0.27	0.28	0.30	0.31	0.34	0.36	0.37	0.30
	Meizhou	0.21	0.20	0.21	0.23	0.26	0.27	0.30	0.33	0.32	0.31	0.27
	Heyuan	0.23	0.22	0.22	0.23	0.25	0.26	0.29	0.30	0.30	0.32	0.26
	Yunfu	0.13	0.10	0.13	0.15	0.19	0.18	0.19	0.27	0.28	0.28	0.19
	Mean	0.23	0.21	0.22	0.24	0.26	0.26	0.28	0.32	0.33	0.33	-
Mean		0.32	0.31	0.33	0.34	0.35	0.35	0.36	0.39	0.40	0.39	-

Source: based on the authors’ calculation.

**Table 8 ijerph-20-04305-t008:** Classification of coupling of three dimensions in 21 cities in Guangdong.

	10-Year Mean	2010	2019
Coordinated coupling	Guangzhou, Shenzhen, Foshan, Dongguan, Zhuhai	Guangzhou, Shenzhen, Foshan, Dongguan	Guangzhou, Shenzhen, Foshan, Dongguan, Zhuhai
Benign coupling	Huizhou, Zhaoqing, Jiangmen, Zhongshan, Shantou, Zhanjiang, Shaoguan	Zhuhai, Zhongshan, Huizhou, Shantou, Jiangmen, Zhaoqing	Zhongshan, Huizhou, Shantou, Jiangmen, Zhaoqing, Shaoguan, Yangjiang, Zhanjiang, Maoming, Chaozhou, Qingyuan
Lower coupling	Chaozhou, Jieyang, Shanwei, Maoming, Yangjiang, Qingyuan, Meizhou, Heyuan, Yunfu	Shaoguan, Heyuan, Meizhou, Shanwei, Yangjiang, Zhanjiang, Maoming, Jieyang, Yunfu, Chaozhou, Qingyuan	Heyuan, Meizhou, Shanwei, Jieyang, Yunfu
Low coupling	-	-	-

Source: based on the authors’ calculation.

**Table 9 ijerph-20-04305-t009:** Classification of coupling coordination of three dimensions in 21 cities in Guangdong.

Category	10-Year Mean	2010	2019
High coordination	-	-	-
Moderate coordination	Guangzhou, Shenzhen, Foshan, Dongguan, Zhuhai, Zhongshan, Huizhou	Guangzhou, Shenzhen, Foshan, Dongguan, Zhuhai, Zhongshan	Guangzhou, Shenzhen, Foshan, Dongguan, Zhuhai, Zhongshan, Huizhou, Jiangmen
Basic coordination	Shantou, Jiangmen, Zhaoqing, Shaoguan, Yangjiang, Zhanjiang, Maoming, Chaozhou, Qingyuan, Heyuan, Meizhou, Shanwei, Jieyang, Yunfu	Huizhou, Shantou, Jiangmen, Zhaoqing, Shaoguan, Yangjiang, Zhanjiang, Maoming, Chaozhou, Qingyuan, Heyuan, Meizhou, Shanwei, Jieyang, Yunfu	Shantou, Zhaoqing, Shaoguan, Yangjiang, Zhanjiang, Maoming, Chaozhou, Qingyuan, Heyuan, Meizhou, Shanwei, Jieyang, Yunfu
Low coordination	-	-	-
No coordination	-	-	-

Source: based on the authors’ calculation.

## Data Availability

All the data used are reflected in the article. If you need other relevant data, please contact the authors.

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
