# Peer review of "Measurement and Coupling Coordination of High-Quality Development in Guangdong Province of China: A Spatiotemporal Analysis"

_ijerph, 2023, doi:10.3390/ijerph20054305_

Round 1

Reviewer 1 Report

There are the following suggestions:
1- The "Discussion" section is not enough and needs more description. Since an important part of the research was done with mental subjector quantification, therefore, in this part, description, comparison, and discussions should be raised that can bring the validity of the research done.
2- The spatio-temporal analysis needs to be investigated in the area of the radius of influence, variography should be done and the environment of the survey should be re-analyzed with the precise determination of the radius of influence, otherwise, due to the mixing of the analysis, the current research is invalid. And it needs a MAJOR revision.

Author Response

Based on the comments and suggestions, we have made careful modifications to the manuscript. The main revisions in this paper and the responses to the reviewers’ comments are appended follow. The detailed information can also be seen in our revised manuscript. Revised parts are marked in red in the revised article.

Reviewer 2 Report

I am grateful for the opportunity to read and review this interesting scientific paper. This is a Chinese case study with a solid methodology that can be published in the International Journal of Environmental Research and Public Health.

However, for this the authors must address improvements:

1. The first would be to add the word 'China' to the title. This is a very China-focused case, and while most readers presumably know where Guangdong Province is, it is convenient to add that it is in China. In the same sense, in the methodology, the study area must be described and it is important to put a location map of the Guangdong province with all its regional units and the location of the cities. On this map, a legend should be added with the population of each city, a very relevant aspect for the issue analyzed in the article. Cities should be represented with circumferences of different sizes depending on their population.

2. In the introduction it is important to add the structure of the article.

3. In all the tables the source must be added, indicating that they are own elaboration if that were the case.

4. Table 1 must add the source of each of the data and the period of time that is studied.

5. In subsection 2.4 Data sources, when line 175 indicates “other data platforms”, these other platforms must be specified. Also when pointing out that there is missing data, it must be clearly stated what it is. What percentage of the total data is missing and has it been interpolated? This is essential for the reliability of the results.

6. In tables 4 and following, the average value for each of the four regions must be added. The analysis compares regional data, but does not appear in the tables. Only the ones from the cities appear.

7. In the discussion section, line 364 alludes to a previous analysis. The bibliographical reference of that previous work should be added.

8. The final text of the discussion section, lines 377 to 384, where the limitations of the study and possible future research are analyzed, should be transferred to the conclusions section.

Author Response

(The authors gave the same response as above.)

Reviewer 3 Report

Thank you for inviting me as a reviewer for the paper titled Measurement and Coupling Coordination of High-quality Development in Guangdong Province: A spatiotemporal analysis.

In general, the paper is well organized. This study analyzes the high-quality development of the economic, social, and ecological environment from 2010 to 2019 in Guangdong. This analysis was performed using the Entropy-TOPSIS model. The strength of this paper is in the problem that is solved, that is, in the case study. The methodology applied in this paper is well-known in the literature.

The authors need to consider the following major points as a limitation or further scope for refining the paper:

1) Literature analysis should be of better quality. The literature mentioned in the introduction should be analyzed in more detail. Also, expand the literature review with new sources (period 2021-2023), such as: Tutak, M., & Brodny, J. (2022). Evaluating differences in the Level of Working Conditions between the European Union Member States using TOPSIS method. Decision Making: Applications in Management and Engineering, 5(2), 1-29. https://doi.org/10.31181/dmame0305102022t; Kizielewicz, B., WiÄ™ckowski, J., Shekhovtsov, A., WÄ…tróbski, J., DepczyÅ„ski, R., & SaÅ‚abun, W. (2021). Study towards the time-based MCDA ranking analysis – a supplier selection case study. Facta Universitatis, Series: Mechanical Engineering, 19(3), 381-399. doi:https://doi.org/10.22190/FUME210130048K.

Based on literature analysis author should better highlight the objective of their paper and to what extent it contributes to close a gap in the existing literature and/or practice. What is the innovative value of the contribution proposed by the authors? This is an essential part of the Introduction and Literature analysis section.

2) The paper's other sections should be discussed in one paragraph at the end of the introduction.

3) Symbols that appear for the first time in equations should be explained, either before or after the equation where they first appear.

4) A flowchart of the methodology is missing.

5) The paper lacks a sensitivity analysis. I propose to perform a sensitivity analysis by applying a change in the weighting coefficients of the criteria. This is important, especially considering that the Entropy method defines the weight coefficients. The obtained results should be discussed.

6) Managerial implications should be considered in a separate section.

7) Show in detail the limitations of the proposed methodology and this study.

8) The authors need to clearly provide several solid future research directions.

Please, mark the requested changes, in the corrected version, in a different color.

Author Response

(The authors gave the same response as above.)

Reviewer 4 Report

The reviewed article prefaces a very interesting study of 3D coupling in Guandong Province is also an example of the application of The entropy weight TOPSIS method. The method was correctly applied and yielded interesting results, so the article deserves to be published. To enhance its relevance and future citation, I suggest some additions and clarifications, which I list in the paragraphs:
Firstly, in the introduction as well as in the conclusion, it should be more explicitly stated why such an analysis serves and how it can be used, what it implies for the practice of development planning and monitoring. A broader discussion of the literature in this regard is certainly needed here, which should go beyond the context of China and justify the adoption of an otherwise interesting method of study somewhat more broadly. Secondly, it is not sufficiently clear whether the data analysed refer only to cities (prefecture capitals) - as indicated in the text - or to entire prefectures as if this can be read from the accompanying maps. There is some lack of reflection on the impact of the list of indicators adopted on the final result of the analysis.
Congratulations on an interesting article.

Author Response

(The authors gave the same response as above.)

Round 2

Reviewer 2 Report

The authors have made a good revision of the article based on my suggestions and have coherently reasoned this revision. For my part, I consider that the article can be published.

Reviewer 3 Report

All the reviewers' comments have been addressed carefully and sufficiently. The revisions are rational from my point of view. I think the current version of the paper can be accepted.